# Realization of Rapid Large-Size 3D Printing Based on Full-Color Powder-Based 3DP Technique

**DOI:** 10.3390/molecules25092037

**Published:** 2020-04-27

**Authors:** Guangxue Chen, Xiaochun Wang, Haozhi Chen, Chen Chen

**Affiliations:** 1State Key Laboratory of Pulp and Paper Engineering, South China University of Technology, Guangzhou 510640, China; 2Guangzhou Financial Service Innovation and Risk Management Research Base, South China University of Technology, Guangzhou 510640, China

**Keywords:** powder-based 3D printing, large-size 3D printing, printing time, cutting-bonding frame

## Abstract

The powder-based 3DP (3D printing) technique has developed rapidly in creative and customized industries on account of it’s uniqueness, such as low energy consumption, cheap consumables, and non-existent exhaust emissions. Moreover, it could actualize full-color 3D printing. However, the printing time and size are both in need of upgrade using ready printers, especially for large-size 3D printing objects. Given the above issues, the effects of height and monolayer area on printing time were explored and the quantitative relationship was given in this paper conducted on the specimens with a certain gradient. On this basis, an XYX rotation method was proposed to minimize the printing time. The mechanical tests were conducted with three impregnation types as well as seven printing angles and combined with the characterization of surface structure based on the scanning electron microscope (SEM) digital images to explore the optimum parameters of cutting-bonding frame (CBF) applied to powder-based 3D printing. Then, four adhesives were compared in terms of the width of bonded gap and chromatic aberration. The results revealed that ColorBond impregnated specimens showed excellent mechanical properties which reached maximum when printed at 45° to Z axis, and α-cyanoacrylate is the most suitable adhesive to bond full-color powder-based models. Finally, an operation technological process was summarized to realize the rapid manufacturing of large-size full-color 3D printed objects.

## 1. Introduction

Three-dimensional (3D) printing is a subversive technology which applies to a variety of materials, such as paper [1,2], powder [3,4,5], and liquid [6]. Based on the principle of additive manufacturing (AM), the technology has the power to greatly reduce the manufacturing time of small- and medium-scale customized products [7], and the majority of 3D printers are equipped with a recycling system to reuse the leftover material in printing process. With the technological innovation and academic research of 3D printing technology in latest 30 years [8,9,10], some typical 3D printing processes are getting mature and have evoked extensive industrial use in areas including bio-medical, aerospace, and culture industries [11,12,13], such as stereo lithigraphy apparatus (SLA) [14], laminated object manufacturing (LOM) [2,15], selective laser sintering (SLS) [16,17], fused deposition modeling (FDM) [18,19], selective laser melting (SLM) [16,20], and three dimensional printing and gluing (3DP) [21,22]. The ASTM (American Society of Testing Materials) F2792 standard further classify these techniques as vat photopolymerization, powder bed fusion (PBF), binder jetting, material jetting, sheet lamination, and material extrusion, directed energy deposition (DED). A method to realize large-size, full-color, and high-precision 3D printing is the focus of research at this stage to widen the scope of this technology, such as military or educational uses (like maps) and artistic or cultural uses (like sculptures and restoration of heritage).

The 3DP technique, whose main printing material is powder [23], was first developed and patented by the Massachusetts Institute of Technology [24], and full-color 3D objects printed by the technique with gypsum or ceramic powder have gained ground in custom models such as medical organization structure [25], portrait, and prototype. Currently, there are two generally accepted full-colour 3D printing techniques similar to 3DP: paper-based 3D printing (LOM) and polyjet technique, which can produce rich gradual colors based on the principle of four-color printing. However, the speed and accuracy of paper-based 3D printing is the main constraint on commercialization, and inadequate penetration of printing ink brings about inaccurate color reproduction on sides. At the current stage in full-color 3D-printing development, the polyjet technique demonstrates desirable printing precision, but compared with the full-color 3DP technique, the restrictions including the enormous amount of required support material as well as expensive material price reduce the manufacturability of large-size 3D printed objects [26].

Along with technical evolution of 3D printing, the size of 3D printers has also increased rapidly, verifying the rising demand for large-size 3D printed objects. Some giant 3D printers have been developed and applied to the construction industry. The Italian engineer Enrico Dini produced a modular machine named D-Shape inspired by the Z-Corp 3D printer by swapping the powder build material with sand or gravel and scaling up the process as well as the machine [27]. Concrete deposition (CD) is one of the emerging technologies of recent years. In fact, the layering and deposition techniques of it resemble FDM [28], i.e., with the help of a programmable logic controller (PLC), the concrete nozzle moved along the outline to deposit the mortar as per slice of digital model [29]. Some buildings with complex structures were printed by these 3D printers, nevertheless, the layer thickness of most 3D concrete printed objects was above 10 mm, so the resolution of CD technique is too low to meet the requirement of most commercial customization models.

Therefore, powder-based 3D printing is found to be a viable choice to manufacture full-color 3D models with large size and high precision. In addition, a number of materials have been developed and will be applied to printing high-precision objects [30,31]. However, the size of printed model is restricted to that of the build bin of a 3D printer, and the newest full-color powder-based 3D Printer (3D Systems ProJet 860 Pro), of which the largest capacity is usually limited in 508 × 381 × 229 mm, still cannot satisfy the sizes of all large-scale models. Moreover, some models with high complexity and precision are prone to fracture in the process of printing or post-processing. A cutting-bonding frame (CBF) for paper-based 3D printing was proposed by Yuan et al. [2] in which one model is 3D printed in parts first and bonded together in post-processing. Another restriction of large-size 3D printing, reducing lengthy printing time, has been investigated in recent years [32]. This can be summarized in three methods. The most immediate way is to upgrade the equipment, such as increasing the number of nozzles and the sliding speed. However, based on an appointed 3D printer, one method involves algorithms that were modified to minimize the trajectory planning [33]. The other method is to increase slice thickness [34], but this method of sacrificing precision is only suitable for some models with simple structure.

However, it is important to note that the aforementioned studies ignored the effect of different machining parameters (like orientation) on printing time, and are based on large-size powder-based 3D printing, many properties related to CBF, such as mechanical properties of cut components and color reproduction accuracy of bonded objects, have not been studied. The present study briefly describes the technological process of powder-based 3D printing as well as an effective post-processing method. Then a quantitative relationship for predicting printing time and the effective method to reduce printing time was developed based on the analyses of two factors in printing time. Finally, the tensile and flexural performances, consistency of surface color, and characterization of microstructure were analyzed to explore the optimum parameters applied in CBF. Taken together, the tests and methods promote the development of large-size 3D printing, as well as reduce the molding time and ultimate construction of a composite structure.

## 2. Results and Discussion

### 2.1. Effects of Z-Axial Height and Monolayer Area on Printing Time

The digital model files were imported into 3DPrint software, and placed in upper-right corner of the build bin after adjusting orientation. Each specimen was printed in a single printing cycle. Moreover, the printing time was automatically recorded by the software and displayed in a pop-up window when the printing cycle was completed (Table 1).

In Exp. A, a positive linear trend between printing time and Z-axial height is shortly indicated from the statistical values, in addition, based on the least square method, the quantitative relationship of cuboids with basal area of 100 cm^2^ can be expressed by Equation (1).
(1)f(x)=32.25x+2.4
where *f(x)* is the printing time (min), and *x* is the Z-axial height (cm).

Based on the relationship between printing time and Z-axial height, the validity of fore mentioned cumulative method (i.e., monolayer printing time can be obtained by dividing the measured data by the total number of layers) was confirmed. In Exp. B, the printing time of the cuboid and cylinder with same basal area is pretty much the same, but that of the cylinder is slightly longer. Figure 1 presents the fitting time-area curve, quadratic polynomial equation, R^2^ (linearly dependent coefficient) and standard error (RMSE) of two groups. That R^2^ is greater than 0.99 and tends to 1 indicate a high degree of agreement between the test data and the fitting function. Getting the printing time of all figures accurately is hard before the printing process because of the diversity of slice shapes, so the time–area curve of cylinders was chosen to roughly predict the printing time of single layer. The specimens with a height of 2 cm were divided into 196 layers obtained from the parameter display. Therefore, Equation (2) was drawn to calculate monolayer printing time.
(2)$$f(x)=\frac{-2.5e^{-5}x^{2}+0.1539x+55.1}{196}$$
where *f(x)* is the printing time (min), and *x* is the graphic area of a slice (cm^2^).

Therefore, prior to printing 3D objects with the same area per layer, the entire printing time can be calculated directly by Equation (3).
(3)$$f(x)=\frac{\left(-2.5e^{-5}x^{2}+0.1539x+55.1\right)h}{2}$$
where *f(x)* is the printing time (min), *x* is the basal area (cm^2^), and *h* is the Z-axial height (cm).

In addition, the predicted printing time of other 3D objects can also be received as the following steps: corresponding digital model is imported into slicing software first, then the parts that need to be sprayed on a slice are extracted from the exported monolayer JPG image, and the total area of sprayed parts of all slice is calculated and substituted in Equation (2).

### 2.2. Effects of Orientation on Printing Time

#### 2.2.1. XYX Rotation Method

Z-axial height of objects is the predominant influence factor of printing time. Moreover, the growth rate of printing time descends gradually as the rise of single-layer area (Figure 1). Hence the most efficient way to reduce the time in 3D printing procedure for a finished digital model is to adjust the orientation in build bin to minimize the Z-axial height. So, we propose the XYX rotation method to find the best orientation quickly.

The function of displaying the Z-axial height is attached in many 3D modeling software. 3ds Max software (Autodesk Ltd, San Rafael, CA, USA) is chosen in this paper and the preset coordinate system is shown in Figure 5b. However, it is unmanageable to catch the orientation with the minimum value of Z-axial height by free rotation in software, thus XYX rotation method was proposed and summarized as follows:(a)The change of Z-axial height value is observed when the digital model is rotated around X axis, and then the orientation with minimum z-axial height is found out in the angle range (0–180°).(b)Based on the orientation of digital model in step 1, it is rotated around Y axis, and obtaining the orientation with minimum Z-axial height in the angle range (0–180°).(c)Based on the orientation of digital model in step 2, repeating step 1.

#### 2.2.2. Verification and Analysis

Ten different digital specimens of three categories were devised to verify the practicability of XYX rotation method. T1 indicates the printing time with the orientation in Table 2, and T2 expresses the printing time of specimens directed by XYX rotation method, therefore the percentages of diminished time to original printing time were counted and demonstrated. Among the time saved of the selected samples, the maximum even reached 65.7% compared to initial printing time. Moreover, when the space of build bin is filled with printing parts (44322 cm^3^), a single printing cycle will occupy more than 40 h. Hence, prior to large-size 3D printing procedure, it is essential to optimize the orientation of each part by the XYX rotation method.

### 2.3. Effects of Impregnants on Mechanical Behavior

The flexural and tensile tests in this paper were conducted with an electronic material testing machine (Instron 5565, Figure 2a). Fixing a reasonable support span of 40 mm in this test, along with inputting the loading rate of 1 mm per minute, then the stress–strain relationship curve was drawn automatically. The mean of flexural modulus of three specimens as well as maximum load data in each group were calculated as shown in Figure 2d. Two impregnating methods, in a manner, increased the bending resistance, and the increment was almost one order of magnitude in ColorBond group, furthermore, the location as well as extending direction of the cracks (Figure 2b) and the fracture processes (Figure 2c) were different in three groups. The first letter (B, C, or E) represents test groups, and the number after the dash represents the specimen number nearest to average.

In the blank group, bending stress mainly depends on the bond of the color binder between layers, but the cohesion between unbonded powder particles or between the bonded block structures is insignificant. When the bottom of specimens failed, the upper layers would not fracture simultaneously, so the stress-strain curve would slowly decline until approaching 0. The crack deviated slightly from the middle in virtue of rough surface of the unimpregnated specimen, and the beginning may be a large pore or a defect around the pressure direction. The added ColorBond, which acts as a binder because it contains a certain amount of cyanoacrylate, filled most of the pores between the particles, where the stress is chiefly dependent on the adhesion of dried impregnant. Once the maximum load was reached, the specimen would abruptly break from the middle by strong tension on the base line, along with the crack would extend apace.

In the epoxy resin group, there is an obvious yield segment in the stress-strain curve, where the coating on the bottom was being rent, after which the curve would drop slowly. Meanwhile, the extending direction of the crack is similar to that in blank group, so the bending stress in the group mainly comes from the adhesion of surface coating, and it is demonstrated that two-component epoxy resin hardly penetrates into powder-based 3D printed specimens.

With the comparison of flexural tests in three groups, despite the fact that coating epoxy resin to the surface of models after impregnation with ColorBond is the best way to strengthen powder-based models, in practice, chromatic aberration will exist on the surface due to the non-uniformity of manual coating and the different impregnated depth (Figure 2b). In addition, the transparency will decrease as coating thickness increases. Thus, the ColorBond impregnating method is the most suitable for full-color models.

### 2.4. Mechanical Tests Based on Different Angles

Despite the great clamping force, pneumatic grips would destroy the specimens directly, and therefore the common grips (CAT. NO. 2710-101) were chosen (Figure 3a). Before running each test, two grips should be aligned to prevent shearing, and the most suitable distance between them was 30 mm based on the size of specimens. During the tensile test, setting the loading rate to 1 mm per minute, and negating some specimens that broke at the nips because the fracture did not necessarily occur under tension; the parameters of flexural tests were similar to that in Section 2.3. Figure 3b,d present separately the stress-strain relationship curves closest to the average for each angle in tensile test and flexural test, and the corresponding average elastic modulus and maximum load were shown in Figure 3c,e.

The distinction among specimens at different angles lies in the number of layers as well as the area of each slice, and as total number of slices increased, the slice area would become smaller (Figure 4). The surfaces of unimpregnated specimens, which are not perpendicular to Z axis, could be seen clear laminated structures similar to the texture on sides of a book. However, surface laminated structures have almost been filled up after impregnation, but there are still mechanical property differences in these specimens at different angles. In tensile test, the stress–strain curves of T0 to T45 fluctuated slightly on account of the sliding caused by small surface roughness under insufficient clamping force, however with the increase of printing angle, especially more than 60°, the friction required to stop sliding was satisfied. The largest tensile strength was detected in T45, and total tensile test was divided into two stages based on the inflection point as follows:

The tensile strength is positively correlated with angle from 0° to 45°. It is found that the stress is chiefly dependent on the adhesion of ColorBond from the analysis in flexural test of different impregnating methods. The more a printed specimen was sliced, the greater the roughness and porosity on side were, and without the barrier of color binder, the impregnant could penetrate into specimens easily, so the tensile modulus and the maximum load increased therewith.

The tensile strength and the maximum load are inversely correlated with angle from 45° to 90°, and the strain of T60 to T90 decreased sharply as Figure 3b. Despite the ample impregnation, the interlaminar gaps of small thickness test specimens are equal to the cracks, which reduced the stability and even induced bending phenomenon when some specimens were printed at 90°.

In the flexural test, there is little difference the flexural strength at each test angle and the regularity is similar to that in tensile test, however the flexural modulus of F0 and F45 is obviously superior to others, which indicates that F0 is the most stable under the bending load, despite the relatively less impregnant in its interior.

Combined with above measured data and analyses, the optimum cutting angle is 45° to Z axis for dividing the digital model prior to the printing process of large-size models, so that the tensile and flexural strength of each impregnated part is maximum, which is also beneficial to the application of high viscosity adhesive. Further, cutting along the direction in Z axis is an effective alternative for some models with small z-axial height and large bottom surface, such as sand tables or topographic maps.

### 2.5. Related Performance Research of Adhesives

The reason why seamless adhesion plays an important part in CBF is that powder-based full-color 3D printing is mainly applied to produce artistic and cultural objects. Apart from the different mechanical properties, the larger width of bonded gap also would darken the appearance of color to observers. The widths of specimens bonded by all four adhesives are less than monolayer thickness (Table 3), which meets the standard of seamless adhesion. However, the mechanical properties of bonded assemblies vary greatly from adhesive to adhesive. Two environment-friendly adhesives, vegetable glue and sodium silicate, are not as suitable for powder-based specimens as organic synthetic adhesives. the fracture of specimens bonded by α-cyanoacrylate was not at the joint in tensile test, and the tensile modulus as well as flexural strength are even higher than the average measured value of non-cutting models. During the bonding process, adhesive would inevitably change the color around the joint, and the correlation between ΔE and visual perception could be roughly summarized in three intervals: the threshold of human eyes to perceive chromatic aberration is 1. 5 NBS and the chromatic aberration can be distinguished when it ups to 3 NBS [35]. In contrast to the color measured values before and after coating, the ΔE in the test groups of double component epoxy resin as well as α-cyanoacrylate belong to the second interval and close to 1. 5 NBS. However, the chromatic aberration is obvious in other groups. Altogether, α-cyanoacrylate is the best adhesive for powder-based seamless adhesion techniques.

## 3. Materials and Methods

### 3.1. Materials and Equipment

As for full-color 3DP technique, the major constituent of powder is usually one material that can be formed rapidly with good molding properties, such as gypsum, ceramic, or quartz. The main materials used in printing process (i.e., VisiJet PXL powder and color binders) were provided by 3D Systems Ltd (New York, NY, USA). A computer installed with 3DPrint software (3D Systems, New York, NY, USA) was selected to operate ProJet 860 Pro 3D printer (3D Systems, New York, NY, USA).

The popular file formats of 3D digital models are STL, WRL and OBJ, while WRL format is commonly used in full-color 3D printing. Prior to the molding process, the 3D digital models are sliced with the operational software (3DPrint, 3D Systems Ltd., New York, NY, USA) which includes the most appropriate slicing method for 3DP technique (Figure 5a). The monolayer thickness range of ProJet 860 Pro 3D printer is set to 0.089–0.102 mm for adjusting the printing resolution, and the default value of monolayer thickness is 0.1016 mm. Each information layer of the sliced model is transmitted by the computer to the 3D printer, in which the nozzle will jet color binders along the images.

In order to maintain equipment at peak performance, all nozzles and slide rails are thoroughly cleaned and the color binders as well as cleaning fluid are replaced with new ones. In addition, the temperature of 12.7–23.9 °C and the humidity of 20%–55% in laboratory are recommended by the machine use guide. Figure 5b exhibits the structure of ProJet 860 Pro 3D printer, in which the feed bin is designed to ensure that powder can fleetly replenish to build bin using the roller, and the molding process can be divided into the following steps in build bin:(a)The powder in feed bin is transported to build bin by the roller to build a substrate with smooth surface;(b)After paving a layer of powder on the substrate, the color binders are sprayed by nozzles onto the powder along X axis;(c)Repeating step b) until 3D models are formed;(d)Impregnation with epoxy resin for post-processing.

Figure 5c presents the process of blowing off excess powder using an air wand. For almost all 3D printing techniques, post-processing is a key step at present to increase surface gloss, eliminate staircase effect and optimize color reproduction, and the fundamental post-processing technics including drying, blowing, abrading and impregnating have been successfully applied to powder-based 3D printing. The printer chamber is allowed to sit for about 30 min for full solidifying of binders, and then excess powder is inhaled by the vacuum hose along the profile of components. Printed parts are transferred to the cleaning chamber to remove residual powder using air wands, brushes and picks. Although Z-Corp Inc. (acquired by 3D systems in 2012) recommends that the parts should be lightly sanded with 220 or 320 grade sandpaper to even out inconsistencies [36], this method is not suggested in practice due to the fact that inappropriate friction will weaken surface color reproduction. The impregnating process is the most crucial during post-processing that some commercial impregnants are recommended in 3D printing industry, as it, in addition to contributing to mechanical properties, also enhances color saturation to ensure the required mechanical, surface, and color attributions [22].

### 3.2. Methods

#### 3.2.1. Estimation and Reduction of Printing Time

The printing time of 3D objects chiefly correlates with three factors, i.e., Z-axial height of printing parts, area per slice, and position in build bin. However, this study investigated large-size model, and in view of the restriction of printing materials and time, the space of build bin should be used to the full, therefore, the time difference due to altered position can be ignored. In addition, it is improbable to print a monolayer model and measure the time, so the trials were conducted with specimens of appropriate height, and the monolayer printing time was obtained by dividing the measured data by the total number of layers.

The color of binder shows few effects on printing time on the principle of that white parts were also be sprayed with transparent binder, and thus the surface color of specimens was entirely mapped with neutral gray. A set of cuboids with a fixed basal area of 100 cm^2^, a square with side length of 10 cm, were printed to test the relationship between printing time and Z-axial height of models. The heights of ten cubes were equidistantly picked from 2 cm to 20 cm owing to the limitation (229 mm) of the height of the build bin. The nozzle slipped from the right side on slide rail to the left end of printing part in inkjet process, hence even for two models with the same monolayer area, the printing time of the one with longer length in the direction of X axis is longer than that of the other. Taking into account the above effect, the printing time measuring trials with different area parameters were divided into two groups, whose specimens were respectively designed as ten cuboids and cylinders with the same height (2 cm), and the basal areas of them, although different in shape, were all set from 100 to 1000 cm^2^ with an interval of 100 cm^2^.

#### 3.2.2. Enhancement of Mechanical Properties

Type of adhesive, secant position and orientation of 3D object are involved in CBF application of oversize specimens. On this basis, mechanical tests during pre- and post-bonding were conducted with a series of powder-based 3D-printed angle-gradient models, besides, the changed mechanical properties and surface color over addition of binders and impregnants were discussed in the following.

Prior to mechanical testing, it is significant to choose an impregnation method with simple operation and excellent optimization in terms of surface color as well as mechanical properties. Therefore, we used the specification in “standard test method for flexural strength of advanced ceramics at elevated temperatures” (ASTM C1211-2002(2008)). The three-point flexural tests were conducted with nine 80 × 15 × 5 mm cuboids divided into three groups. One group is the blank control group, in which specimens were not impregnated, and two common impregnants, ColorBond (3D Systems) and two-component epoxy resin (Ergo), which can effectively improve surface color reproduction accuracy [37], were severally applied to the other groups. The surface of powder-based specimens should be fully immersed for 20 s in ColorBond group, while coated by soft brush in epoxy resin group. All specimens were laid using the XYX rotation method.

Additive manufacturing brings about orthotropic surface structure and mechanical properties, which should be taken into account when evaluating tensile strength or flexural strength found on CBF.

The particularity of a 3D printing material and molding process restricts the applicability of the cylinders with small radius to tensile test, so cuboids impregnated by ColorBond were designed for mechanical testing with the dimensions of 80 × 15 × 5 mm, and seven printing angles—namely 0°, 15°, 30°, 45°, 60°, 75°, and 90°—defined as the angle between the longest edge and the horizontal plane were set in tensile and flexural test groups. Hence, Z-axial height is the minimum at 0° and the maximum at 90°. The specimens were named T0 to T90 for tensile tests and F0 to F90 for flexural tests. In addition, the tests of each angle were conducted with three specimens printed in different printing cycles to reduce the effect of printer performance, and corresponding printed models were designated, e.g., T0-1, T0-2, and T0-3.

#### 3.2.3. Selection of Adhesives

The digital model of T45 was divided equally into two cuboids with the size of 40 × 15 × 5 mm, and bonding experiments were conducted with four kinds of adhesives. Double component epoxy resin (Ergo) and α-cyanoacrylate (Deli) organic synthetic adhesives are excellent impregnants in themselves [37], hence they are very suitable to bond the powder-based specimens; vegetable glue (WingArt) is one of organic natural adhesives with abundant sources, low price and environmental protection; inorganic adhesives such as sodium silicate have been widely used in construction and model manufacturing industries, and liquid sodium silicate was picked with 3.3 M and 40 Baume in this experiment. Each adhesive was evenly coated with 20 mg on the splitted surface to fabricate three bonded assemblies per test group, after standing about 10 h, the bonded gap of each specimen was observed by metallographic microscope and its width was measured on random three points. Then the tensile and flexural tests were carried out separately. A monochrome plate (R = G = B = 128, RGB is a staple color mode which represents red, green, and blue channels, and the color intensity span was prescribed as 0–255) with the surface area of 100 mm × 100 mm was printed and divided into four parts, and a spectrophotometer (X·Rite-i1 pro2) was employed to measure the surface colors five times (at four corners as well as the center) on each part, then four adhesives were coated separately, and the surface colors were measured in the same position after drying. Finally, the chromatic aberration formula was adopted to calculate ΔE_2000_ (CIEDE 2000) [38].

## 4. Conclusions

This paper studied the rapid manufacturing of 3D models based on inkjet 3DP technique to break through the capacity constraint of printer with cutting-bonding frame. The correlation between printing time, Z-axial height, and basal area was explored with preset gradient parameters, and a simple formula for estimating printing time was presented, along with the XYX rotation method was proposed to minimize printing time by changing the orientation of the digital model. With respect to the optimization characteristics of impregnants and the orthogonal anisotropy of 3D printed objects, the tensile and flexural tests were conducted with specimens impregnated by two impregnants and printed at seven different angles. In addition, the mechanical properties and the surface color variation before and after the gluing process of assemblies bonded by four adhesives were compared under the standard of seamless adhesion.

The results showed that the increment of Z-axial height is the predominant factor affecting the printing time compared with the expansion of basal area. Ten universal test models taken from three categories verified the effectiveness of XYX rotation method, in which the maximum shortened time reached 65.7% of original print time. The mechanical properties of powder-based 3D printed objects were substantially enhanced through the impregnating process, and the impregnating adequacy of ColorBond impregnant as well as the printing stability at different angles resulted in the inconsistency of mechanical properties expressed in SEM images and test data, from which 45° to Z axis is the optimal cutting-bonding angle for powder-based 3D printing based on CBF. α-cyanoacrylate adhesive is the best choice for seamless adhesion due to the narrowest bonded gap, excellent mechanical properties and trifling effect on surface color. Finally, a systematic large-size 3D printing operating process was summarized.

On the basis of the above results, a realizable procedure for the large-size powder-based inkjet 3DP technique was summarized as follows and could be extended to other 3D printing technics:(a)Digital models are placed with the XYX rotation method.(b)Models are split into several parts at an angle of 45° to Z axis according to the size and the capacity of build bin.(c)Following the printing and de-powdering processes, all parts are impregnated by ColorBond for about 20 s.(d)The impregnated parts are bonded by α-cyanoacrylate adhesive with the sizing of 25 mg/cm^2^.

Most powder materials can be applied to inkjet 3DP technique combined with suitable binders and auxiliaries due to the simple molding process. With the parameters and methods suggested in this work, large-size models with useful mechanical properties can be printed by diverse materials using the inkjet 3DP technique and applied in more fields.

## Figures and Tables

**Figure 1 molecules-25-02037-f001:**
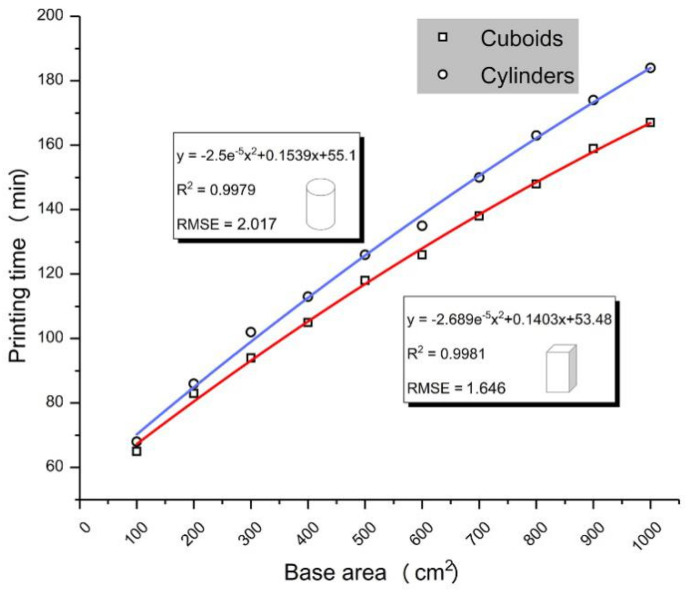
Printing time - basal area fitting curves.

**Figure 2 molecules-25-02037-f002:**
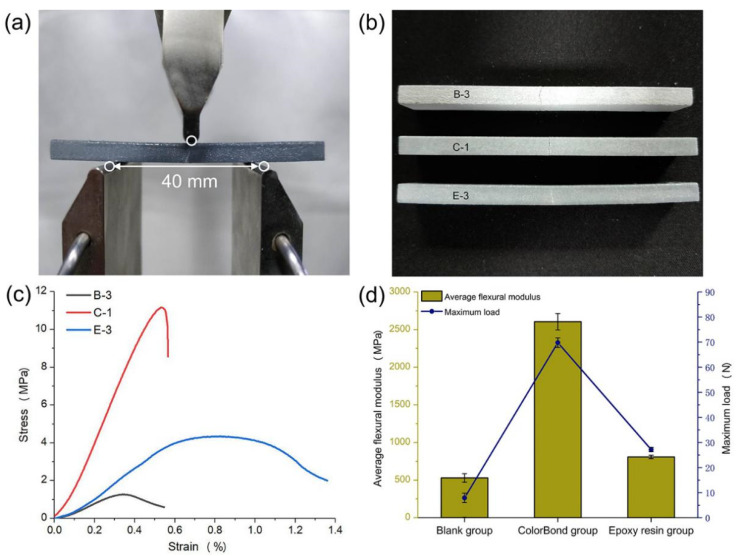
Flexural tests of specimens subjected to three impregnating process. (**a**) Flexural testing device; (**b**) Image of flexural failure specimens; (**c**) Stress-strain curves of three-point flexural test; (**d**) Bending-related properties measurements.

**Figure 3 molecules-25-02037-f003:**
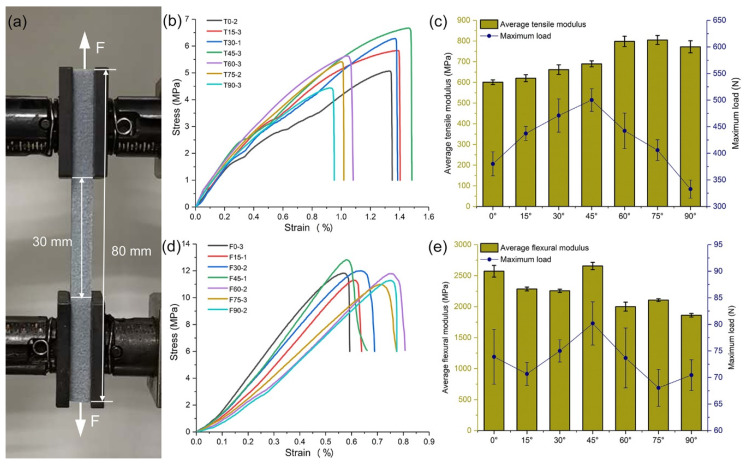
Mechanical tests at different printing angles: (**a**) Image of a uniaxial force stretched powder-based specimen; (**b**) Tensile stress-strain curves of a series of specimens; (**c**) Tensile-related properties measurements; (**d**) Stress-strain curves of three-point flexural test; (**e**) Bending-related properties measurements.

**Figure 4 molecules-25-02037-f004:**
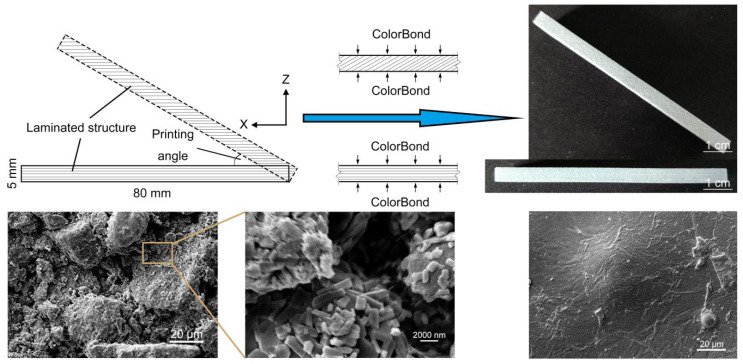
Dissimilarities of laminated structures and permeating process of the specimens printed at different angles, and characterization of surface appearance before and after impregnation.

**Figure 5 molecules-25-02037-f005:**
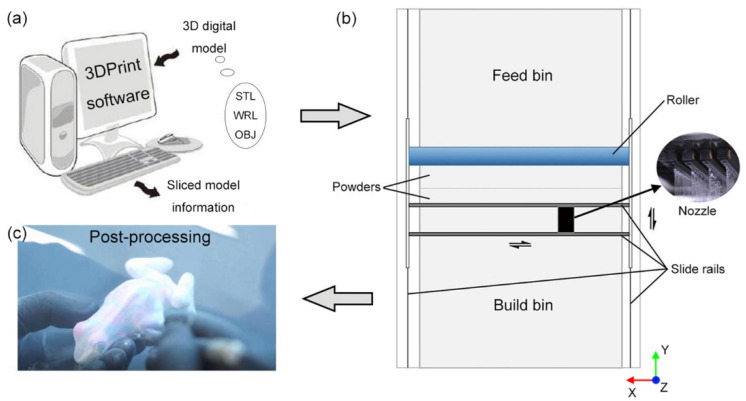
Workflow of powder-based 3DP technique: (**a**) Slicing and transmission of digital model; (**b**) Printer structure chart and coordinate system; (**c**) Post-processing.

**Table 1 molecules-25-02037-t001:** Selected parameters and obtained printing time of specimens.

Exp. A (Effects of Z-Axial Height on Printing Time)				Exp. B (Effects of Monolayer Area on Printing Time)							
No.	Parameters			Z-Axial Height (cm)	Basal Area (cm^2^)	Cuboids			Cylinders		
	Basal Area (cm^2^)	Z-Axial Height (cm)	Printing Time (min)			No.	Side Length (cm)	Time (min)	No.	Radius (cm)	Time (min)
A1	100	2	65	2	100	B11	10.00	65	B21	5.64	68
A2		4	131		200	B12	14.14	83	B22	7.98	86
A3		6	196		300	B13	17.32	94	B23	9.77	102
A4		8	262		400	B14	20.00	105	B24	11.28	113
A5		10	327		500	B15	22.36	118	B25	12.62	126
A6		12	392		600	B16	24.49	126	B26	13.82	135
A7		14	452		700	B17	26.46	138	B27	14.93	150
A8		16	517		800	B18	28.28	148	B28	15.96	163
A9		18	582		900	B19	30.00	159	B29	16.93	174
A10		20	648		1000	B10	31.62	167	B20	17.84	184

**Table 2 molecules-25-02037-t002:** Measurements and differences of printing time of ten random 3D models.

Specimens	Standard Primitives			Compound Objects			Body Objects			
	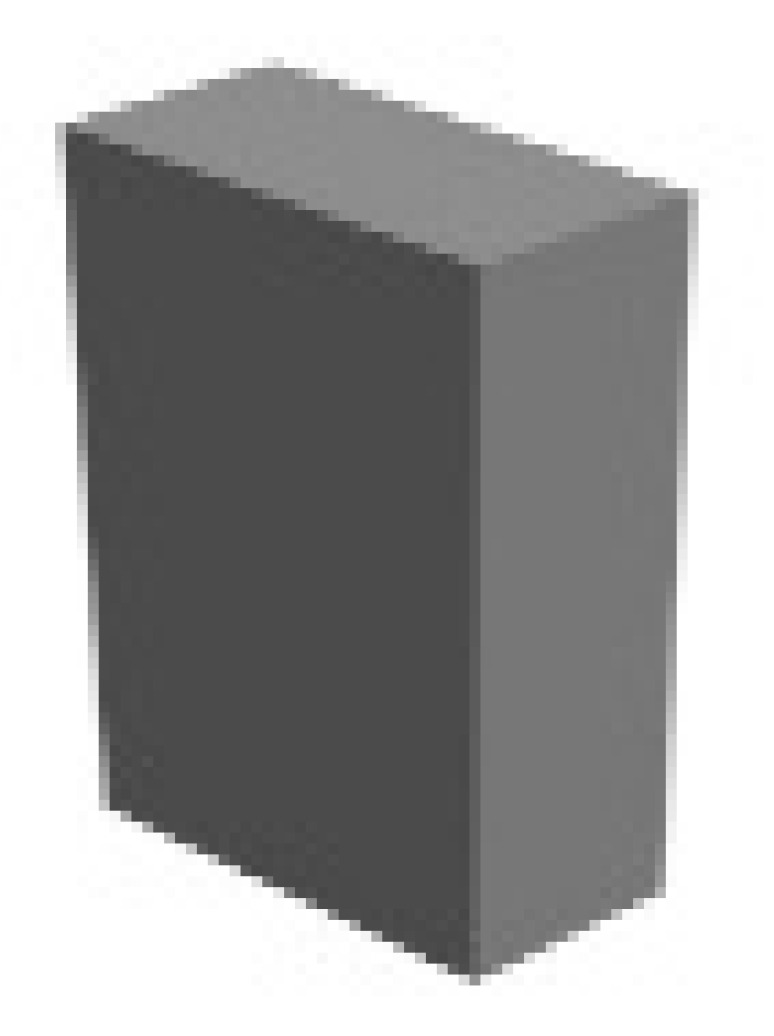	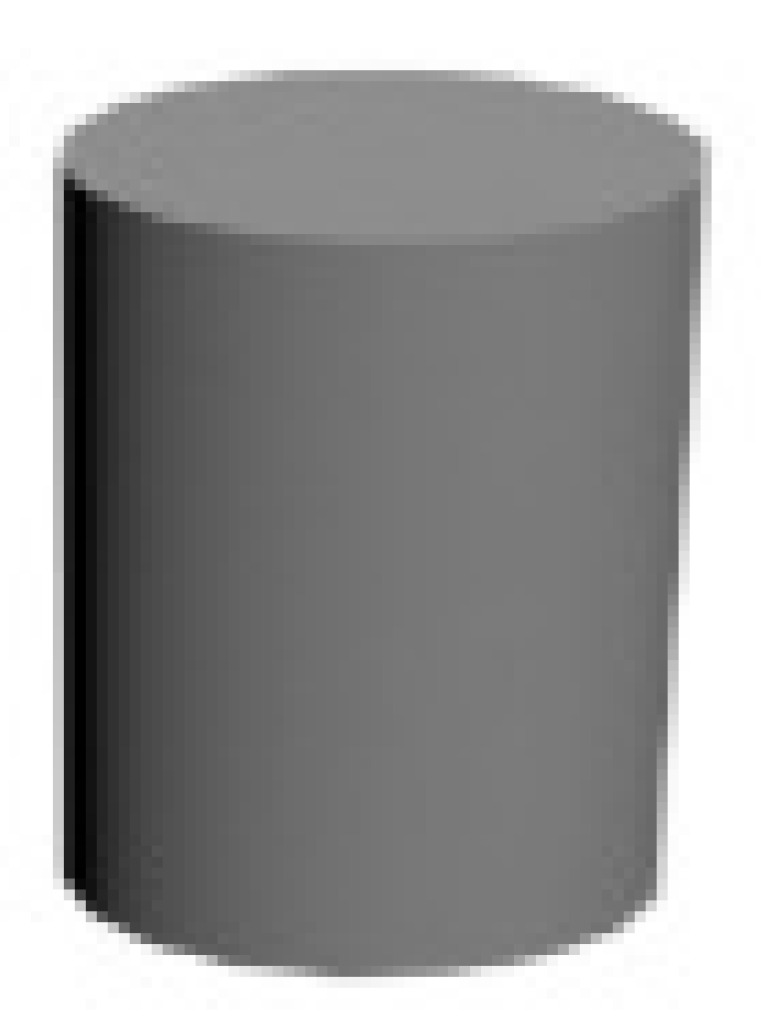	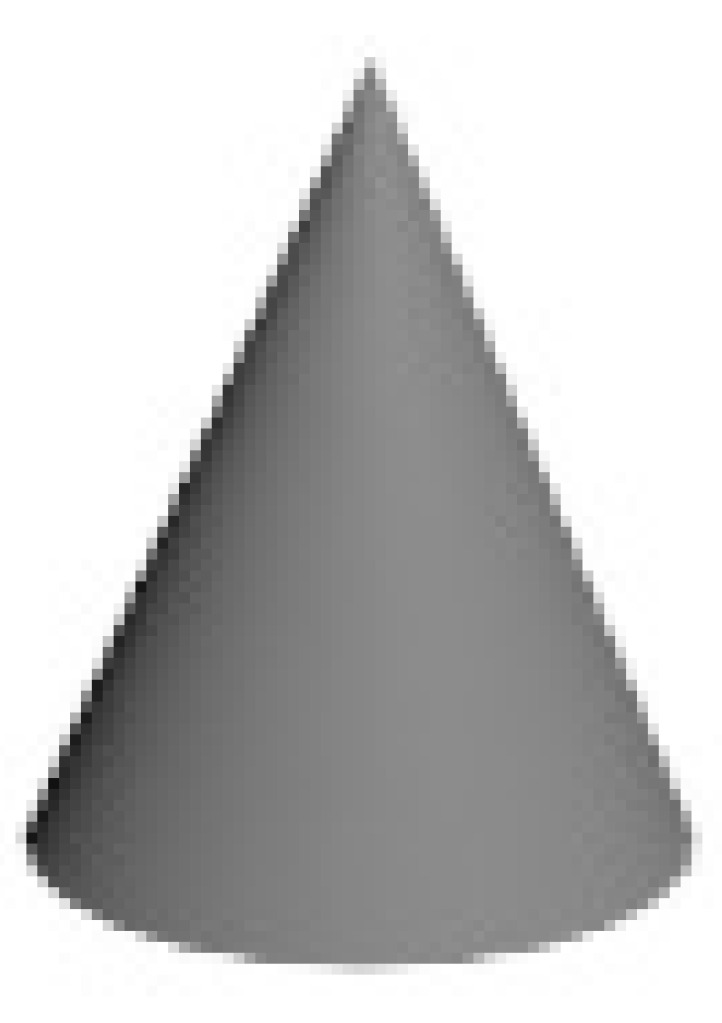	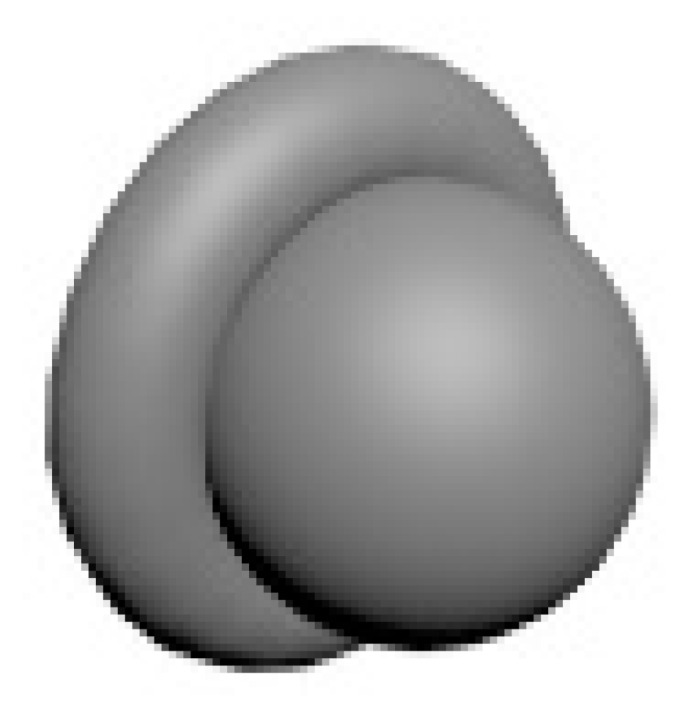	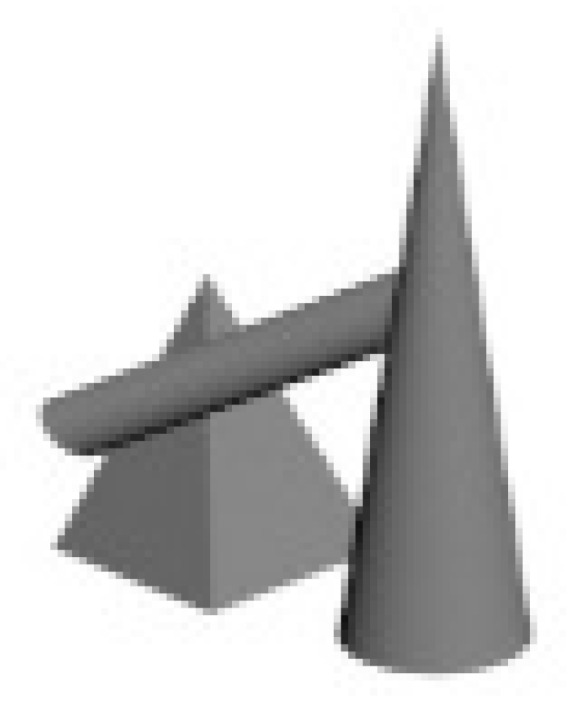	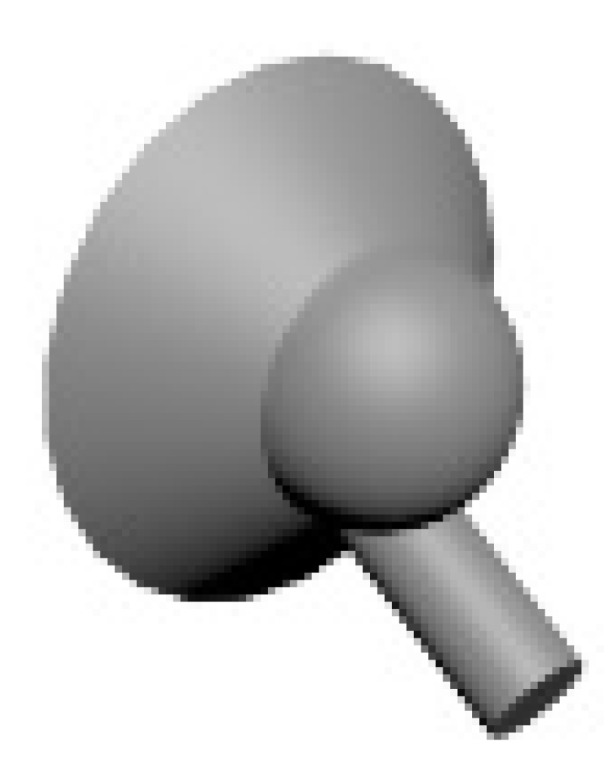	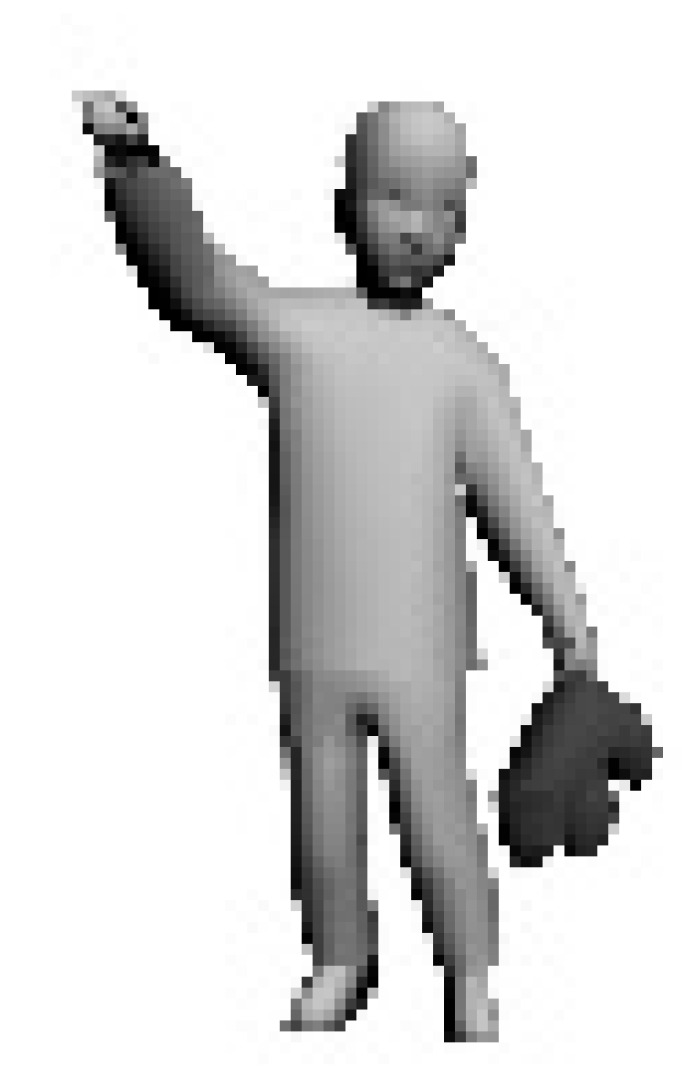	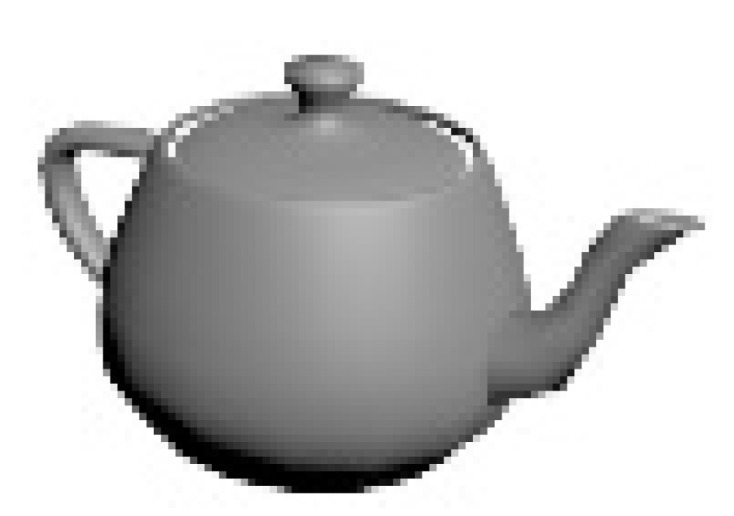	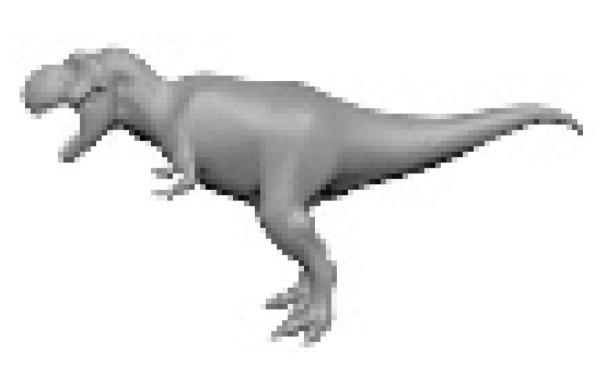	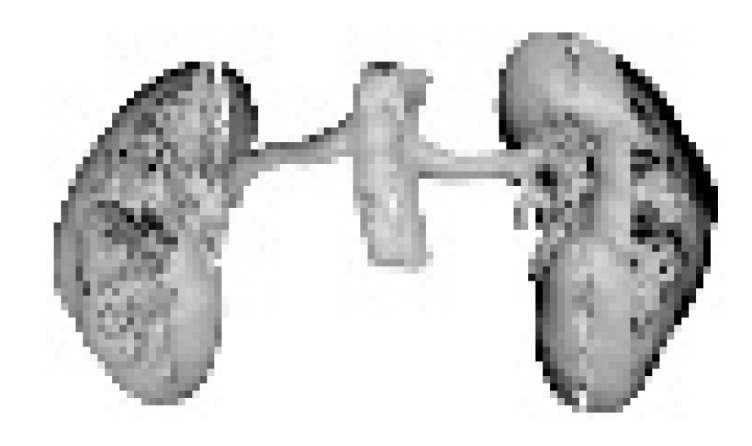
T1 (min)	630	628	554	743	534	656	547	384	576	337
T2 (min)	223	434	320	602	183	508	316	377	382	251
(T1-T2)/T1 (%)	64.6	30.9	42.2	19.0	65.7	22.6	42.2	1.8	33.7	25.5

**Table 3 molecules-25-02037-t003:** Parameter values related to seamless adhesion of four adhesives.

Adhesives	Width of Bonded Gap (μm)	Tensile Test			Flexural Test		Chromatic Aberration (ΔE, NBS)
		Tensile Strength (MPa)	Tensile Modulus (MPa)	Fracture Location	Flexural Strength (MPa)	Flexural Modulus (MPa)	
Vegetable glue	52	1.43	317.58	A	0.53	430.52	5.52
Sodium silicate	22.7	1.16	209.53	A	0.68	477.92	3.98
Double component epoxy resin	24.7	4.64	598.55	A	8.48	1786.74	1.78
α-cyanoacrylate	15.3	6.58	786.33	N	13.57	2052.22	1.57

A represents the joint; N represents the location apart from the joint of bonded assembly.

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
