# Peer review of "Realization of Rapid Large-Size 3D Printing Based on Full-Color Powder-Based 3DP Technique"

_molecules, 2020, doi:10.3390/molecules25092037_

Round 1
Reviewer 1 Report
This is my review of “Realization of Rapid Large-size 3D Printing Based on 2 Full-color Powder-based 3DP Technique” by Chen et al. I recommend this paper be accepted with major revisions for English language and usage. The paper presents a study of how to optimize 3D printing parameters to result in superior mechanical properties.
The authors need to state in the introduction why people would want to 3D print large, color models. I can think of educational uses (like maps), but what are other industrial and scientific uses?
The authors should talk about impregnation with epoxy during their introduction of the workflow (Figure 1).
There are numerous run-on sentences. For example, 194-196 should be split at “hence.” Lines 251-55 should be three sentences, splitting at “moreover” and “hence.” These words, along with therefore, should not be used to join two sentences into one. In most cases where they use a semicolon, they should just end the previous sentence and start a new. Semicolons are rare in current English usage.
In 238-246, the authors should explain why this particular polynomial expression fits their data. What is the underlying physical basis for this equation fitting their data?
The authors should use the word “laminated” to describe their models rather than “striped.” Striped connotes a surface feature, while “laminated” connotes through-going layering.

Author Response
Response to reviewer
Dear Reviewer:
Thank you for your comments concerning our manuscript entitled “Realization of Rapid Large-size 3D Printing Based on Full-color Powder-based 3DP Technique” (ID: 775568). We all appreciate your detailed work. Those comments are all valuable and very helpful for revising and improving our paper, as well as the important guiding significance to our researches. We have studied your comments carefully and have made correction which we hope meet with approval. Revised portion are marked in red in the paper. The main corrections in the paper and the responds to the reviewer’s comments are as flowing:
Responds to the reviewer’s comments:
Reviewer #1:
1. Response to comment: The authors need to state in the introduction why people would want to 3D print large, color models. I can think of educational uses (like maps), but what are other industrial and scientific uses?
Response: We have re-written this part according to the Reviewer’s suggestion. Besides, there are many applications of large-size color 3D printing, such as large size handicraft manufacturing, cultural relic restoration, mould manufacturing, etc.
2. Response to comment:The authors should talk about impregnation with epoxy during their introduction of the work-flow(Figure 1).
Response: Thanks for your suggestion. We have re-written the work-flow that included the impregnation process with epoxy. Besides, related impregnation methods were described in detail in previous studies (Ref. 36). In this paper, we also described the operation of epoxy resin (Lines 170-172) and the problems in the experimental results (Lines 295-298).
3. Response to comment:There are numerous run-on sentences. For example, 194-196 should be split at “” Lines 251-55 should be three sentences, splitting at “moreover” and “hence.” These words, along with therefore, should not be used to join two sentences into one. In most cases where they use a semicolon, they should just end the previous sentence and start a new. Semicolons are rare in current English usage.
Response: We are very sorry for our incorrect writing and have made correction according to the Reviewer’s comments. Thank you very much for your suggestions on language and structure changes.
4. Response to comment:In 238-246, the authors should explain why this particular polynomial expression fits their data. What is the underlying physical basis for this equation fitting their data?
Response: We are very sorry for our negligence that forget to explain the meaning of R2 in the data. The underlying physical basis for this equation fitting their data is least square method.
5. Response to comment:The authors should use the word “laminated” to describe their models rather than “” Striped connotes a surface feature, while “laminated” connotes through-going layering.
Response: We have made correction according to the Reviewer’s comments.
Once again, thank you very much for your comments and suggestions.

Reviewer 2 Report
The research program investigates the process and post-processing parameters relating to 3DSystems' binder jetting technology. The experimental plan was designed to extend the use of binder jetting technology to the production of parts having dimensions greater than the working volume of the system considered. However, no large parts have actually been produced or studied; no results refer to large parts.
Therefore the title does not focus on the research topic.
The title should reflect the object of the work: since large parts have not been produced nor investigated, this title is not consistent.
In the text, the authors use commercial terminology that is not always updated or precise; they classify 6 additive technologies while this technologies are classified in 7. It would be appropriate to use the classification proposed in the international standards (ASTM and ISO): binder jetting instead of 3DP (from row 36 to row 39)
The term ployjet (perhaps polyjet!) should not be confused with FDM (lines 47 and 48) because they are two different technologies both capable of producing colored parts.
Bibliographical references are extensive but hardly focused on the topic of the investigation.
Concerning Materials and methods
Paragraph 2.1: the binder jetting technology may include a post-processing phase of hardening by firing but the authors do not declare whether they performed or avoided it before the experimental testing on the samples.
Section 2.2: inserting images of the print layout of cuboids and cylinders will help in understanding the reproducibility of the results.
Paragraph 2.2.1: the XYX method has been investigated for different types of objects but each one starting from only one initial configuration. The study could be generalised by applying the method XYX starting from a random orientation of parts.
The results of the investigation relating to mechanical properties are particularly interesting and extensive, although authors do not deepen the mechanisms (i.e. chemical-physical) that determine them.
Some questions would remain open:
- is there a correspondence between the performances declared by the manufacturer, the estimate of the printing time by the printer management software and the effective printing speed determined by the authors?
- why has the XYX method of part orientation not been compared with other optimization methods or software available on the market in order to generalise the results of the study and justify its use for large-size parts printing?
- it would be advisable to verify the applicability of the proposed method (lines from 383 to 390) to actual large-size part.
Author Response
Response to reviewer
Dear Reviewer:
Thank you for your comments concerning our manuscript entitled “Realization of Rapid Large-size 3D Printing Based on Full-color Powder-based 3DP Technique” (ID: 775568). Those comments are all very valuable and helpful for revising and improving our paper, as well as the important guiding significance to our researches. We have studied your comments carefully and have made correction which we hope meet with approval. Revised portion are marked in red in the paper. The main corrections in the paper and the responds to the reviewer’s comments are as flowing:
Reviewer #2:
1. Response to comment:Therefore the title does not focus on the research topic.The title should reflect the object of the work: since large parts have not been produced nor investigated, this title is not consistent.
Response: This is a very good comment. Considering the Reviewer’s suggestion, we have discussed and thought that the concept of large-size is difficult to define. In the end, we defined large-size as the size over the build bin. Therefore, the series of research conducted by the CBF framework used in this article is consistent with this definition.
2. Response to comment:In the text, the authors use commercial terminology that is not always updated or precise; they classify 6 additive technologies while this technologies are classified in 7. It would be appropriate to use the classification proposed in the international standards (ASTM and ISO): binder jetting instead of 3DP (from row 36 to row 39)
Response: Thanks for pointing out the problem. ASTM has proposed seven classifications of 3D printing, we have provided additional explanations in the article (from row 43 to row 46).
3. Response to comment:The term ployjet (perhaps polyjet!) should not be confused with FDM (lines 47 and 48) because they are two different technologies both capable of producing colored parts.
Response: We are very sorry for our incorrect writing and have made correction according to the Reviewer’s comments.
4. Response to comment:the binder jetting technology may include a post-processing phase of hardening by firing but the authors do not declare whether they performed or avoided it before the experimental testing on the samples.
Response: The models in this article are solidify directly with the binder and impregnants, so they are not suitable for this post-processing
5. Response to comment:the XYX method has been investigated for different types of objects but each one starting from only one initial configuration. The study could be generalizedby applying the method XYX starting from a random orientation of parts.
Response: Thanks for the valuable advice. The XYX rotation method is suitable for different orientations. An initial orientation was selected in the experiment to calculate specific values in printing time. More experiments of random orientation of parts will be our main work in the future.
Some questions would remain open:
6. Response to comment:Is there a correspondence between the performances declared by the manufacturer, the estimate of the printing time by the printer management software and the effective printing speed determined by the authors?
Response: This is actually an interesting question and worthy of further study. In the case of the device we used, we found that the actual printing time is longer than estimated. In particular, the error between actual and estimated time after XYX method optimization is larger than before. In other words, the error of estimated time is more relevant to the XY plane than the Z height. We haven't looked for the exact cause, but we suspect that it may depend on the condition of each equipment. For example, if the slide track of X-axis is stained with powder and not lubricated in time, the movement time of each print will be longer than estimated, which may cause a large error after superposition.
7. Response to comment:why has the XYX method of part orientation not been compared with other optimization methods or software available on the market in order to generalise the results of the study and justify its use for large-size parts printing?
Response: It is really true as Reviewer suggested. This is the direction of our further research to achieve optimization over existing methods or software. It is necessary to optimize the printing time and mechanical properties (such as brittleness value) of the 3D printing model.
8. Response to comment:it would be advisable to verify the applicability of the proposed method (lines from 383 to 390) to actual large-size part.
Response: Thanks for the instructive proposal, we have applied this method to military 3D maps and achieved good results. However, more complex models still need to be verified in our future work.
Special thanks to you for your good comments.
We tried our best to improve the manuscript and made some changes in the manuscript. These changes have been marked in red in the revised paper.
We appreciate for Editors/Reviewers’ warm work earnestly, and hope that the correction will meet with approval.
